# Retrodicting the rise, spread, and fall of large-scale states in the Old World

**James S. Bennett** [ID] *

School of Oceanography, University of Washington, Seattle, Washington, United States of America

* jsb11@uw.edu

## Abstract

Understanding the rise, spread, and fall of large-scale states in the ancient world has occupied thinkers for millennia. However, no comprehensive mechanistic model of state dynamics based on their insights has emerged, leaving it difficult to evaluate empirically or quantitatively the different explanations offered. Here I present a spatially- and temporally-resolved agent-based model incorporating several hypotheses about the behavior of large-scale (>200 thousand $km^2$) agrarian states and steppe nomadic confederations in Afro-Eurasia between the late Bronze and the end of the Medieval era (1500 BCE to 1500 CE). The model tracks the spread of agrarian states as they expand, conquer the territory of other states or are themselves conquered, and, occasionally, collapse. To accurately retrodict the historical record, several key contingent regional technological advances in state military and agricultural efficiencies are identified. Modifying the location, scale, and timing of these contingent developments allows quantitative investigation of historically-plausible alternative trajectories of state growth, spread, and fragmentation, while demonstrating the operation and limits of the model. Under nominal assumptions, the rapid yet staggered increase of agrarian state sizes across Eurasia after 600 BCE occurs in response to intense military pressure from 'mirror' steppe nomadic confederations. Nevertheless, in spite of various technological advances throughout the period, the modeled creation and spread of new agrarian states is a fundamental consequence of state collapse and internal civil wars triggered by rising 'demographic-structural' pressures that occur when state territorial growth is checked yet (warrior elite) population growth continues. Together the model's underlying mechanisms substantially account for the number of states, their duration, location, spread rate, overall occupied area, and total population size for three thousand years.

**Data Availability Statement:** Code and data are available on zenodo: https://doi.org/10.5281/zenodo.5748186.

**Funding:** The author received no specific funding for this work.

## Introduction

Understanding the rise, spread, and fall of large-scale states in the ancient world has occupied thinkers for millennia [1–10]. These historians and others have narratively explored many detailed, context-specific explanations for different states, focusing on the proximal causes for a state's expansion and collapse; the Roman Empire, famously, has a catalog of over 200 conjectured contributory causal factors to its fall [11]. Explanations have generally fallen into one

**Competing interests:** The author declares that no competing interests exist.

of two camps, one focused on the internal structures—demographics, division of labor, political institutions, state finances—that drive rulers and other state agents, and the other on external factors and path dependencies—ecological constraints, the realities of inter-state competition, technological limitations and incentives—to explain why states follow certain trajectories. Each approach seems to explain one part of the story or a subset of cases.

Scholars have despaired (and sometimes disparaged attempts) at finding any systematic, comprehensive macro-historical, social 'forces' and 'laws' or 'mechanisms' that could causally and consistently account for the observed historical trajectories of states, let alone make plausible quantitative estimates and predictions about the dynamics of the scale, duration, or rates of expansion of these states [12–15]. Nevertheless, recent investigations [16, 17] suggest that understanding the state's cooperative ability to make competitive, expansive war [7, 18] and its later collective inability to support the demands of its growing population [19] together, provide a compelling framework for a comprehensive account of state dynamics. Prior modeling based on this 'demographic-structural' conjecture [17] suggested that each state's warrior 'elites' pursuing a common expansive and then divisive military behavior (described below) can account for much of the historical rise, spread, and collapse dynamics of agriculturally-based (agrarian) states in Afro-Eurasia between the late Bronze Age and the end of the Medieval era (1500 BCE to 1500 CE), the 'Old World'. However, in spite of accounting for 60% of the spatial variance of 3000-year agrarian imperial density, this model was unable to account for, among other things, the rapid rise of very large agrarian states (>1 million km$^2$) such as the Persian, Han, Mauryan, and Roman empires and the rise of states in south Asia.

Historians have speculated that the rapid increase of the largest agrarian state sizes during the first millennia BCE could be due to improvements in military technology and organization in response to increasingly powerful extortionary threats from steppe nomads across central Eurasia [20–24]. Turchin [23] conjectured that an observed statistically-significant pairing of large agrarian states with 'mirror' steppe nomadic confederations may be due to an autocatalytic arms race between the two types of polities after the steppe nomads invented horse-mounted cavalry and associated tactics around 1000 BCE. He argued that the agrarian-steppe boundary had become a 'meta-ethnic frontier' in which nomadic peoples were perceived by nearby agrarian states as posing an existential threat due to their different lifestyle, language, religion and military superiority. This created an impetus for radical military reorganization and increased internal cooperation to counteract the perceived threat. Although perhaps triggered by different regional causes, once started, both sides made continual investments in military effectiveness, weapons, and logistics that required ever larger polities on both sides of their ecological divide to support.

Prior models have addressed this prediction differently. Bennett [17] implemented an exogenous and ahistorical doubling of state military efficiency in 700 BCE simultaneously across the Old World agrarian states, but, nevertheless, was unable to account for several very large scale states. Turchin *et al.* [16] pursued an alternative model. Rather than modeling population-driven warfare and state collapse directly, their model simulated the rise and fall of agrarian states via their stochastic acquisition, maintenance, and loss of abstract 'ultra-social' traits. Increased cooperation and state size occurred when an increased number of traits within polities was reinforced and spread through surrounding regions in response to diffusing military prowess from the steppe. Although their model was able to account for up to 65% of the spatial variance of 3000-year agrarian imperial density, their model ahistorically assumed steppe military pressure began in 1500 BCE simultaneously across the steppe, generated thousands of very small, short-lived states unobserved in the historical record to achieve millennial-scale imperial densities, and also failed to account for the sharp increase in historical state sizes after 600 BCE. Neither model predicted the rise or location of nomadic confederations.

Here I present the results of a significant extension of [17] that substantially retrodicts the increasing size and scale of large-scale ($> 200$ thousand $km^2$) polities in the Old World, including the timing of the rise of different 'mirror' states. The model significantly expands the demographic-structural agrarian state model to include steppe nomadic confederations and a simple implementation of Turchin's autocatalytic mechanism to investigate the consequences of increasing military efficiency, state size, and confederation formation on agrarian states after the invention and subsequent spread of nomadic horse cavalry from the Pontic-Caspian steppe region starting in the late 2nd millennium BCE [25]. Analyzing different 'counter-factual histories' (see [10] for a discussion of counter-factual reasoning in historical analysis) generated by varying aspects of the model (i.e., disabling or trying various alternative behavioral mechanisms, varying critical scale parameters, changing exogenous technological contingencies) yields quantitative insights into various key historical events while revealing the model's strengths and shortcomings. Overall the model provides the first fully-articulated quantitative, mechanistic explanation of the bulk statistics and polity dynamics for three thousand years of Afro-Eurasian history.

## Materials and methods

Reduced to its essentials, the model implements a few key behaviors of what I term 'agrarian states' and 'nomadic tribes and confederations' operating as agents interacting in a realistic representation of Old World Eurasian geography and ecology (see **Fig 1**). Tessellated into rectangular regions of about 25 thousand $km^2$ at the equator, each region is marked with a fixed 'biome type' (e.g., agricultural, steppe, desert), has a mean elevation, and is connected to adjacent regions along Cartesian directions, geography permitting.

In addition to geographical constraints, the state agents also respond to a set of exogenous, contingent regional events and technologies (detailed in **S1 File**):

- Where and when initial states arise and their scale

- Where and when various agricultural regions change their productivities

- Where and when horse cavalry was invented and the speed of its diffusion across the steppe

- Where and when long-distance transportation is available at different efficiencies

I investigate the impact of modifying these contingencies on state behavior and spread below.

The model behavior is compared against two large, comprehensive historical datasets: Total and regional yearly population estimates of Afro-Eurasia in this period taken from the 'KK10' dataset of [26] and a regional encoding, every century, of the location of large-scale ($> 200$ thousand $km^2$) polities (both agrarian and nomadic) in the Old World adapted from [16].

### Agrarian states

The agrarian state behavioral model, based on [17], treats a state as a collective of people that cooperate to maintain, exploit, and increase both the total and residual available carrying capacity (the number of people sustainably supportable per $km^2$) under their control. Residual carrying capacity ('opportunity'), in particular, permits the growth of additional population, which, in turn, increases the military ability of the state to expand carrying capacity, subject to certain limits discussed below. Modeling the various coupled internal (demographic and structural) forces and external competitive pressures that change the states' carrying capacity and population provides a framework for investigating the dynamic behavior of states, forces often debated separately by historians.

## A: Relative agricultural intensification before 300 CE

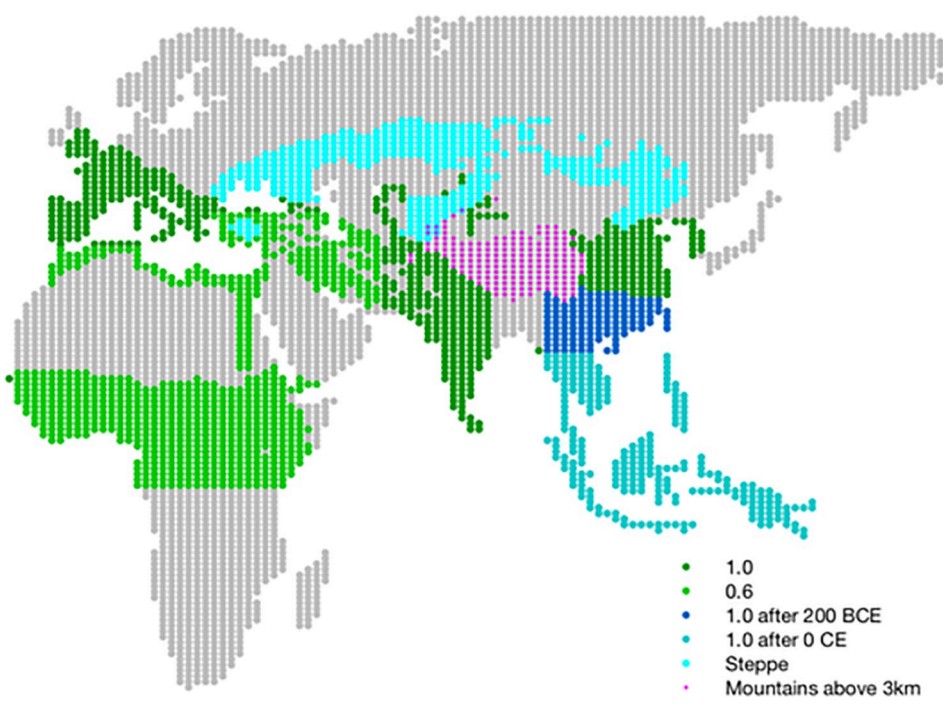

- 1.0
- 0.6
- 1.0 after 200 BCE
- 1.0 after 0 CE
- Steppe
- Mountains above 3km

## B: Relative agricultural intensification after 300 CE

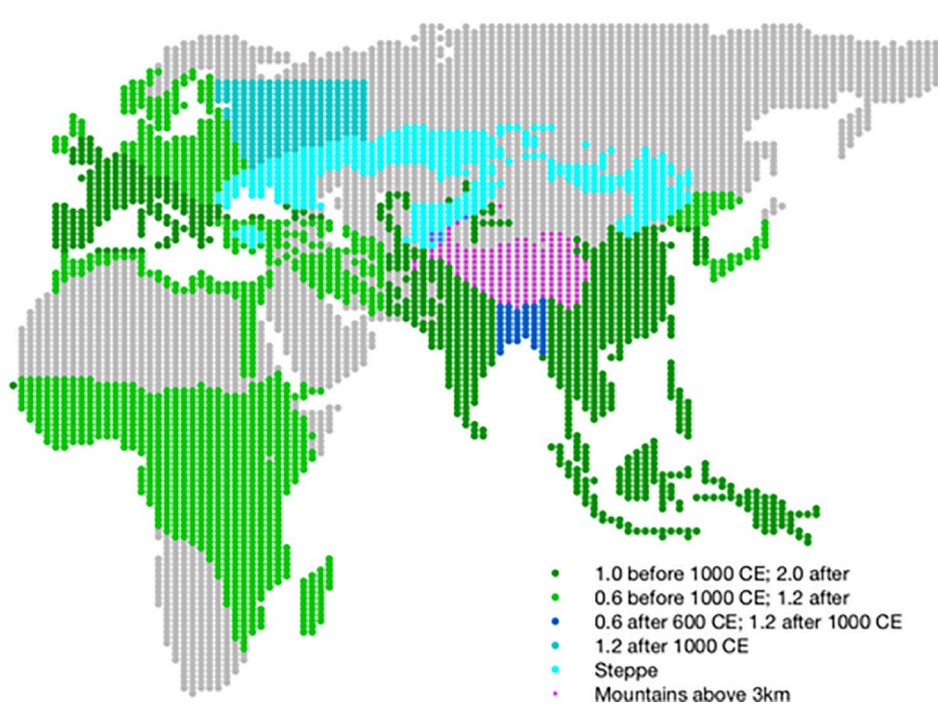

- 1.0 before 1000 CE; 2.0 after
- 0.6 before 1000 CE; 1.2 after
- 0.6 after 600 CE; 1.2 after 1000 CE
- 1.2 after 1000 CE
- Steppe
- Mountains above 3km

**Fig 1. Relative agricultural productivity.** The modeled Afro-Eurasian geography used in the simulations. Relative agricultural productivity due to improved technology available to agrarian states in different eras are highlighted by different colors. Map and biome data republished from [16] under a CC BY license, with permission from Proceedings of the National Academy of Sciences, original copyright 2013.

In the model there are two primary ways for a state to increase its carrying capacity: Annex additional agricultural land (extensification) or improve the productivity of the land annexed (intensification). Annexation reflects competition between states for valuable land. Each annexed region has a local prevailing total carrying capacity and population. Regional population grows logistically up to the region's total carrying capacity assuming a uniform base net natural birthrate of 1.35% per year; the net effective birthrate declines to zero as regional carrying capacity is saturated and the local Malthusian limit is approached. Within a multi-region state, some (typically very small) fraction of the population can migrate per time step to intra-state regions of greater residual carrying capacity (higher 'opportunity'), which has the effect of potentially delaying the exhaustion of local opportunity while increasing the net effective birthrate of those previously more-saturated regions.

Using historical locations of large-scale states provided by [16], Bennett [27] observed that the KK10-estimated Old World population could be accounted for, surprisingly, if the regional carrying capacity provided by the efforts of a multi-region state was three times that of non-state-occupied ('hinterland') agricultural regions. This increase was presumably due to both better defense against predation and increased returns to scale of internal coordination via shared infrastructure, techniques, and rapid adaptation of agricultural 'packages' to new regional environments [28]. Thus, as regions came under the influence of states, the tripling of carrying capacity supported higher populations (and higher effective net birthrates). Bennett [27] found that before 1000 CE, states could apparently support up to, on average, 12 people/ $km^2$ compared with the 4 people/$km^2$ of a single 'hinterland' region. To account for the significant increase in population after 1000 CE, state-based productivity apparently doubled again, to 24 people/ $km^2$ throughout the Old World. Different theaters in the Old World support somewhat different underlying productivities in different eras, largely reflecting differences in climate and soil. The nominal regional agricultural productivity assumptions of the model at different times are summarized in **Fig 1**.

The dynamics of agrarian states follows [17]. Within each modeled agrarian state, the labor of its (regional) population is divided into two specialized sectors: A large number of farmers produce both their sustenance plus a surplus that supports a smaller group of warrior 'elites'. These specialized (and hence more efficient) warriors are responsible, collectively, for defending the state's regions from invasion by other states and, if possible, annexation of additional regions into the state at the expense of their rival political neighbors. The model assumes 5% of a state's regional carrying capacity is reserved to the warriors. Intra-state migration occurs within sectors at potentially different rates but there is no mobility between sectors: People are born to their station and remain there.

The state's ability to project power to distant regions increases with total higher warrior population but decreases in proportion to the length of the state's political border and to increasing logistical support requirements away from the state's single fixed central depot, established when the state is created. The logistical (and hence military) efficiency of a state is summarized by a single parameter, labeled $s$, which scales the number of fighting warriors that can be supported by a supply officer per 100 'effective' km (accounting for transport costs in different terrains, see below). As the effective distance to the border increases, additional supply officers are required to support the existing supply officers, rapidly decreasing (nearly exponentially) the total number of available fighting warriors on political borders [29]. The

scale of historical states and populations in this period suggests that *s* varied between 1 and 3 warriors/supply officer/100 km in this period ([17], S1 File). Here I investigate mechanisms and events that vary this parameter as well as the scale of effective military transport distances some states enjoyed.

States can attack connected, stochastically-chosen agricultural regions along their political border that provide a minimum-expected productivity, subject to an 'attack budget' of no more than 45% of the states' border regions per two-year time step. Annexation occurs when one state can project 20% more warriors into a contested region than the occupying state, reflecting the victor's ability to defend the annexed territory after the exchange.

Annexation is not without cost. In addition to the logistical support costs of projecting power, in an exchange, victorious warriors lose 10% of their (projected) number while the vanquished warriors lose 80% of theirs (empirically estimated). These warriors are removed proportionally according to the regional distribution of warriors throughout their respective states at the time of annexation. Remaining vanquished warriors in the annexed region become warriors for the victorious state. The death of warriors both increases the residual carrying capacity for the warrior segment ('elite opportunity') available to the victorious state while significantly weakening the remaining vanquished state. The marginal increase in carrying capacity is largest and requires the fewest warriors when states annex hinterland, which has low initial carrying capacity and low warrior population.

The territorial growth of a state is checked in several ways. Geographically, regions with low or no expected productivity (high mountains, deserts, steppe) are avoided and strong political rivals can prohibit advance depending on the unfolding dynamics of changing projected warrior power. Indeed, one or more powerful rivals can (incrementally) annex a state in its entirety; thus states can succumb to foreign invasion. Further, even in a large basin of relatively weak hinterland regions, a state can only grow to the logistical limit structurally set by its military efficiency parameter (*s*). Once these different checks apply a state is unable to annex further territory.

Even when expansion is checked, however, the state's population continues to increase logistically toward saturation of the now limited, remaining residual regional carrying capacity. As the warrior population approaches its regional allocated carrying capacity (and in spite of low rates of migration to other state regions), demographic-structural theory [19, 30] suggests that this 'elite overproduction' will generate increasingly fierce internal competition for control of the dwindling and increasingly concentrated spoils of the farmer's productivity. The theory suggests that while remaining cooperative enough to defend the state from inter-state war for a time, their 'intra-elite' competition eventually foments a societal crisis, causing a fragmentation into two or more rivalrous (and internally highly-cooperative) factions, which engage, eventually, in civil war. In the model, the increasing factionalism caused by elite overproduction is approximated by a simple threshold: A civil war is triggered whenever some region's warrior carrying capacity becomes 85% saturated, well before the Malthusian limit. The state is then split stochastically into two new factional states which, by construction, share a new political border. Civil war then proceeds identically to inter-state war and annexation.

While the power of the individual factional states roughly halves, that power is now projected from the separate central depots of each state. Thus new and old territories on the periphery of the parent state come under changed power regimes. Land that could not be annexed by the original state may succumb to a smaller but now closer faction and land that was once well supported by the larger army of the parent is no longer, allowing an external political rival to annex territories away from the factions. This shifting power dynamic, which emerges from the factional rearrangement of state boundaries, overall propels the spread of

more states away from the parent, eventually filling all connected available agricultural land [17].

The agrarian state model thus combines fundamental aspects of the two broad theories of state formation and collapse into a single coherent and causal theory of state operation driven by population growth and, eventually, structural pressures. As all states share identical parameters (except $s$, which changes here based on interactions with other polities as described below), each state (and whatever un-modeled institutions they employ) is assumed to be as competent as the next, where the stochastic choices that drive annexation and faction-formation substitute for whatever flair and folly individual leaders and their entourages may have provided. The emergent dynamic behavior of states is nevertheless complex, path-dependent and contingent, driven by and predictive of state extents, external conflict, internal rivalries and population scales.

In terms of the model, the rapid increase of state sizes after 600 BCE outlined in the Introduction suggests that $s$ roughly doubled in this period. To address the shortcomings of previous models and following [23] the present model employs a simple implementation of an increasingly strong steppe nomadic extortionary threat which slowly diffuses across the steppe, impacting the timing of the rise of very large states in different agricultural basins.

## Nomadic tribes and confederations

Historically, nomadic pastoralists, occupying the central Eurasian steppe shown in **Fig 1**, were organized in loose tribal arrangements [20–22, 24, 31] and engaged in raising livestock, notably horses, and trading with nearby agrarian regions. Initially their ability to demand ('extort') better terms with these states through military force was negligible. However, toward the end of the 2<sup>nd</sup> millennium BCE, certain tribes in the Pontic-Caspian steppe region developed improved political organizations [31] and sophisticated horse cavalry tactics and related technology, notably the saddle and powerful compound bow (see [25] for data on horse cavalry spread throughout Eurasia). Tribes armed with these advantages began raiding and extorting the relatively wealthy, nearby agrarian states. Later, as the wealth of these larger states increased, the nomads confederated their tribes to improve their number and logistical ability to strike deeper into the agrarian states themselves. Agrarian states found the growing nomadic threat difficult to defend against, let alone eliminate, with conventional agrarian armies adapted for annexation: There was little to annex and logistically support agrarian armies on the steppe since, by their nature, nomadic pastoralists had no central settlements to capture and control. The strategic retreat deep into the steppe, a favored tactic of the nomads, thwarted many large agrarian armies. In spite of their increased military power, most large-scale agrarian states eventually employed a variant of the Chinese *heqin (ho-chin)* policy of providing various prestige items (of ever-increasing quality and quantity) and trade concessions to the nomads in order to avoid their extortionary raids altogether.

In the model, these behaviors are approximated as follows. At the start of the simulation the steppe and nearby non-agricultural regions are allocated to a set of roughly 50 stochastically-composed nomadic tribes, each covering about 400 thousand km$^2$. Initially these tribes pose no military or extortionary threat to any nearby agrarian states. At some point, depending on the simulation specification, horse cavalry is invented by the tribes located within a particular longitudinal band in a particular year; the nominal simulations assume the invention occurs in the Pontic-Caspian steppe region (roughly 45˚E) in 1000 BCE. Horse cavalry then spreads to additional tribes at a specified rate, nominally such that all tribes have acquired horse cavalry around 500 years later [25, 31], implying a relatively slow spread and adoption rate of around 10 km/year.

Acquisition of horse cavalry permits a nomadic tribe to extort any agrarian states exposing 10% of their territory within a certain agricultural 'strike zone' logistical distance from the steppe. In the simulations discussed here, the tribal strike zone distance is set to 200 km (determined empirically); see **Fig 2**. In addition, the tribe is perceived by the extorted agrarian states as having an 'effective' $s$ ($s_n$) of 1.4, slightly higher than the initial agrarian state $s$ value of 1.1. Faced with *any* political rival with a larger military efficiency (including, eventually, other agrarian states), modeled agrarian states increase their own $s$ at a rate that depends on the difference in efficiencies. In this way, while the nomadic threat directly impacts nearby agrarian states, eventually states bordering them but outside of the direct range of the nomads will increase their $s$, triggering a cascade of potentially larger states that fill the agrarian lands. Agrarian states formed from demographic-structural crises inherit their parent's military efficiency.

The directly extorted agrarian states, expanding due to their increasing military efficiency and then swelling warrior population, become substantially larger and more populous. To maintain the nomad's ability to extort these larger opportunities a set of modeled independent 'inner' tribes [20] can confederate when the combined population of all the agrarian states any one tribe can extort separately exceeds a threshold. Empirically, a threshold of around 6 million people permits the model to match the timing of the first historical (Scythian) confederation and the subsequent rise of the Achaemenid empire in southwest Asia. The strike zone distance for a confederation also nearly triples to 550 km (see **Fig 2**), potentially increasing the number of states extorted by the confederation. Confederations present higher values of $s_n$ (up to 2.1, nearly double the initial agrarian state value) and continue to grow in size by impressing additional 'outer' tribes not already part of another confederation at a rate around 1 tribe per 19 million additional extortable agrarian people up to a maximum of roughly a half of the total nomadic tribal territory, as suggested by the historical record.

The model assumes nomadic confederations disintegrate into their constituent tribes stochastically every few (25 year) generations mimicking the lateral succession conflicts endemic to steppe nomads [20, 32, 33]. However, new confederations form rapidly due to large nearby agrarian populations. Details about trade between agrarian states and nomadic polities, notably along the Silk Road, are ignored; nomadic population is not tracked. Finally, in the model, neither nomadic polities nor agrarian states invade and control the other. Thus 'ibn Khaldun' agrarian states—such as several north African Islamic states and the historic Tang and Yuan empires, which were invaded and then controlled by nomadic groups (and which later succumb to agrarian 'luxuries' in a sequence that may well be demographic-structural itself), are not simulated in this model ([3]; see [34] for an explanation of this 'Khaldunian cycle').

One emergent model behavior, present in the original model of [17], is that two or more adjacent powerful states, which check each other on their mutual border, will expand away from each other into more easily conquered nearby hinterland, develop power as a consequence, perhaps leading, eventually, to one state dominating the other depending on the relative availability of hinterland. Under the present model this 'reflux' behavior is often both distorted and amplified in response to the asymmetric nomadic (and later agrarian) threat, potentially permitting modeled states that are threatened early to have a 'first-mover' advantage as they and their descendants expand into agricultural basins.

Historically both nomadic confederations and agrarian states continued to improve their military technologies in this period, including adopting heavier armor, improved bows and stirrup systems, and eventually siege engines [31, 35]. The model assumes that the impact of these innovations and any other military changes are reflected as changes in $s$. Interestingly, I find no additional (exogenous) increases in $s$ associated with these historical events beyond the model above that are required to explain the historical data of this study. This suggests

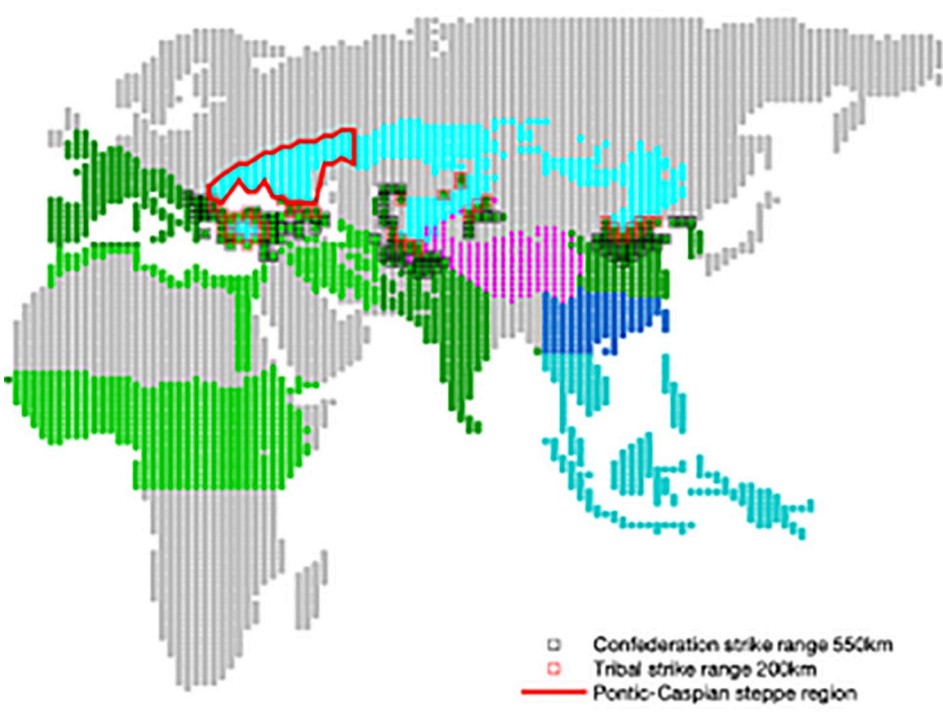

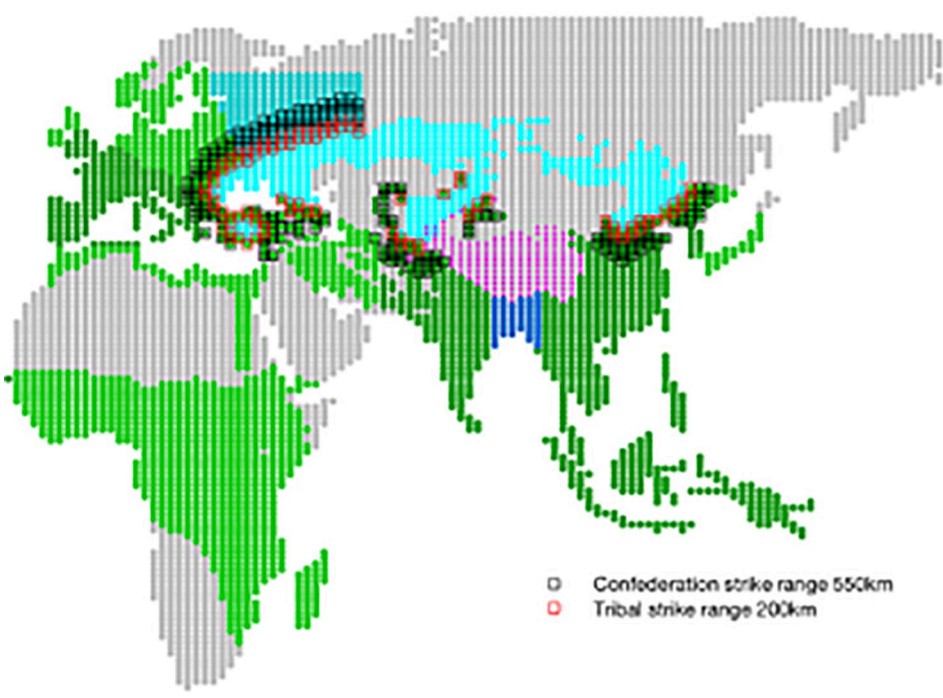

**Fig 2. Nomadic strike zones.** Agricultural regions within nominal strike zones highlighted for nomadic tribes (red, 200 km) and confederations (black, 550 km) in different eras. Agricultural regions and steppe colored as in **Fig 1**. Map and biome data republished from [16] under a CC BY license, with permission from Proceedings of the National Academy of Sciences, original copyright 2013.

whatever asymmetric impact these innovations had between different states was short-lived and served largely to maintain but not increase overall (relative) military efficiency.

## Model operation

Computationally, the behavior of each polity agent, agrarian or nomadic, is governed by internal pressures as well as interaction with its local geographical, political, and technological environment. At the start of each simulation, after the steppe regions are stochastically arranged into nomadic tribes, a few agricultural regions are seeded at different times with small (2–3 region) agrarian states that serve as initial spreading centers, typically in locations and times of historically known states: the Nile valley and the Fertile Crescent in 1500 BCE and later in the middle Yellow River valley in 1200 BCE. Simulations begin in 1500 BCE and unfold independently of history for 3000 years; various composite statistics are then compared to the historical datasets. At each two-year time step, external technological changes, if any, (i.e., changes to agricultural intensification, sea and desert transport improvements, invention and spread of horse cavalry) are made, and then every extant modeled polity evaluates its local threats and opportunities and pursues the behaviors described above, including growing and migrating population, annexing neighboring land, forming confederations, extorting states, and potentially collapsing and forming new polities. An animation of a typical run (see **S1 Video**) and additional, detailed procedural, mathematical and quantitative details of the model, including an enumeration of nominal parameter values and contingencies are available in **S1 File**.

## Results

The overall statistical results from 32 simulations of the model in the Old World using the nominal parameters and contingencies are shown in **Figs 3** through **6** below. **Fig 3** compares the historical and predicted large-scale polity density at 1000-year intervals during the study period. The left panels reflect historical densities from [16]; right panels are model density predictions including both agrarian states and nomadic confederations.

During the last half of first 1000-year interval after 1500 BCE (**Fig 3A**), the initially moderate extortionary pressures of first modeled steppe nomadic tribes with horse cavalry in the Pontic-Caspian steppe region enable modest agrarian state size and population increases in the Near East. A few centuries later the population of these states exceeds 6 million triggering the creation of a simulated nomadic confederation in that region. The confederation's stronger military pressure leads in turn to very large-scale agrarian states in the Iranian plains to the south around 600 BCE. In the model, these states require efficient desert transportation to rapidly attain significant size (see below). As mentioned above, this simulated state dynamics corresponds roughly to the timing and scale of the rise of the historical Achaemenid/Scythian states. However, the simulated states in this period have not expanded quite as far as historical states. In addition, the modeled nomadic confederation(s) also pester the small agrarian states in the Anatolian and Balkan regions.

The modeled slow, 500-year diffusion of horse cavalry across the steppe implies it was just arriving on the east Asian steppe border at end of the first 1000-year interval and no modeled eastern nomadic confederations have yet arisen. Indeed, the still low population of the relatively small east Asian modeled agrarian states seeded there in 1200 BCE means that, in a

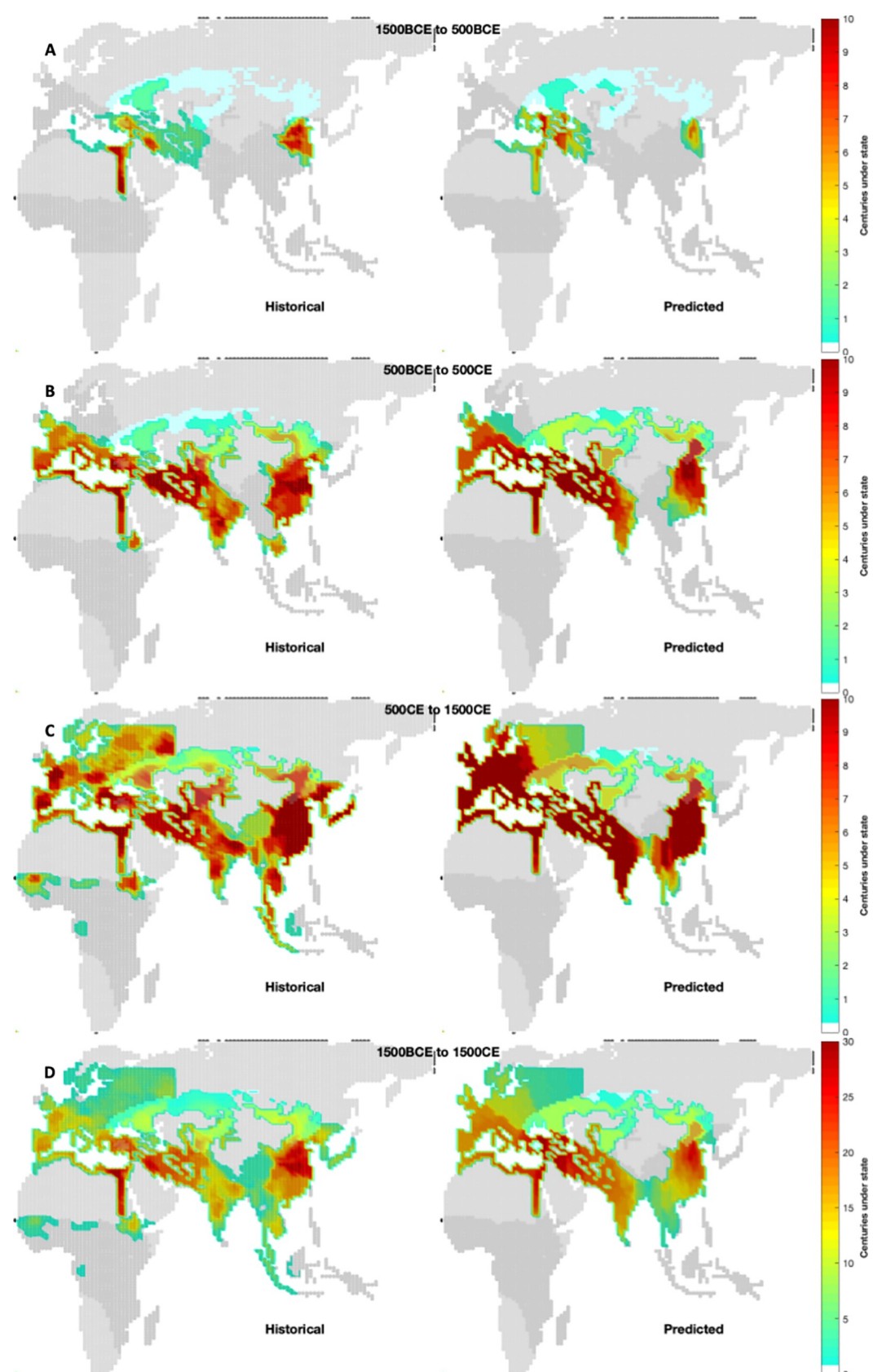

**Fig 3. Predicted polity densities.** Mean large-scale polity density for the Old World at the end of each 1000-year interval (**A**, **B**, and **C**) and over 3000 years (**D**) beginning in 1500 BCE. Left panels reflect historical data from [16] for large-scale states ($\geq$ 10 regions); right panels reflect model predictions. Darker grey indicates available agricultural regions in each period; light blue indicates steppe. Map, biome and historical data republished from [16] under a CC BY license, with permission from Proceedings of the National Academy of Sciences, original copyright 2013.

sequence similar to the southwest Asian one above, the nomadic tribes pester them for several hundred years before the agrarian population grows sufficiently to trigger the formation of a nomadic confederation around 200 BCE, which corresponds roughly to the rise of the historic Han/Xiongnu states.

By the end of the second 1000-year interval in 500 CE (**Fig 3B**), several modeled nomadic confederations and their 'mirror' large-scale agrarian states have risen and fallen across central Eurasia. Modeled nomadic confederations consistently occupy nearly all of the central steppe region in this period as the agrarian states within striking distance have all now grown large and have substantial and stable populations.

During the second 1000-year interval descendants of the large-scale agrarian states from southwest Asia expand into the Indian subcontinent from the northwest, a behavior consistent with history even if the simulation lags in the occupation of south Asia and does not trigger an immediate rise of a large-scale state in the Gangetic plain. This behavior is an improvement over [17], which suggested the occupation of India occurred via east Asian states (unless that model was seeded with an ahistorical large Indus Valley state in 1500 BCE). During this same period, other modeled descendants of the southwest Asian states expand into western Europe, using sea-borne travel (see below), largely along the lip of the Mediterranean but perhaps a bit too far into northern Europe compared with historical dynamics. In the simulations, the expansion into these secondary theaters is accomplished by states that inherit (or face) high agrarian military efficiency but are not, themselves, directly pressured by the steppe nomads.

Finally, in the third interval beginning in 500 CE (**Fig 3C**) additional agricultural land becomes productive in eastern Europe and then northwestern Russia (see **Fig 1B**), allowing nearly all of the agricultural regions in Europe to become occupied by large-scale states. Some of these states pair with nomadic confederations that can now form along this newly exposed strike zone (**Fig 2B**). In east Asia descendant large-scale states expand into the southern Mekong regions from the north in a manner similar to the European and Indian secondary expansion, again, away from direct steppe nomadic pressure, as those lands open to productive agriculture. The historical sub-Saharan states in the west and the Indonesian states in the east do not arise in the simulations both because of the lack of modeled long-distance connections and the expansion time required by the state-forming regions to their north. Overall, however, the model's staggered locations and timing of nomadic confederation and mirrored large-scale agrarian state formation correspond largely to observed historical dynamics across Afro-Eurasia.

**Fig 4** compares several additional composite statistical measures of these simulations against the historical data over time. Overall, the total area occupied (**Fig 4A**) by both agrarian states and nomadic confederations expands at a rate that is very similar, if biased slightly lower during the second 1000-year interval, to the observed, historical rate. The mean expansion rate emerges from the underlying military and demographic-structural behavior of the modeled states from the different spreading centers into their available basins. The early, low historical expansion prior to 600 BCE is well matched driven in the model by geography, the lower initial military efficiency of agrarian states, and lower extortion pressure from nomadic tribes versus nomadic confederations in central Eurasia. The substantial variation in agrarian land area occupation is driven by stochastic choice of battle fronts, slight timing and location differences

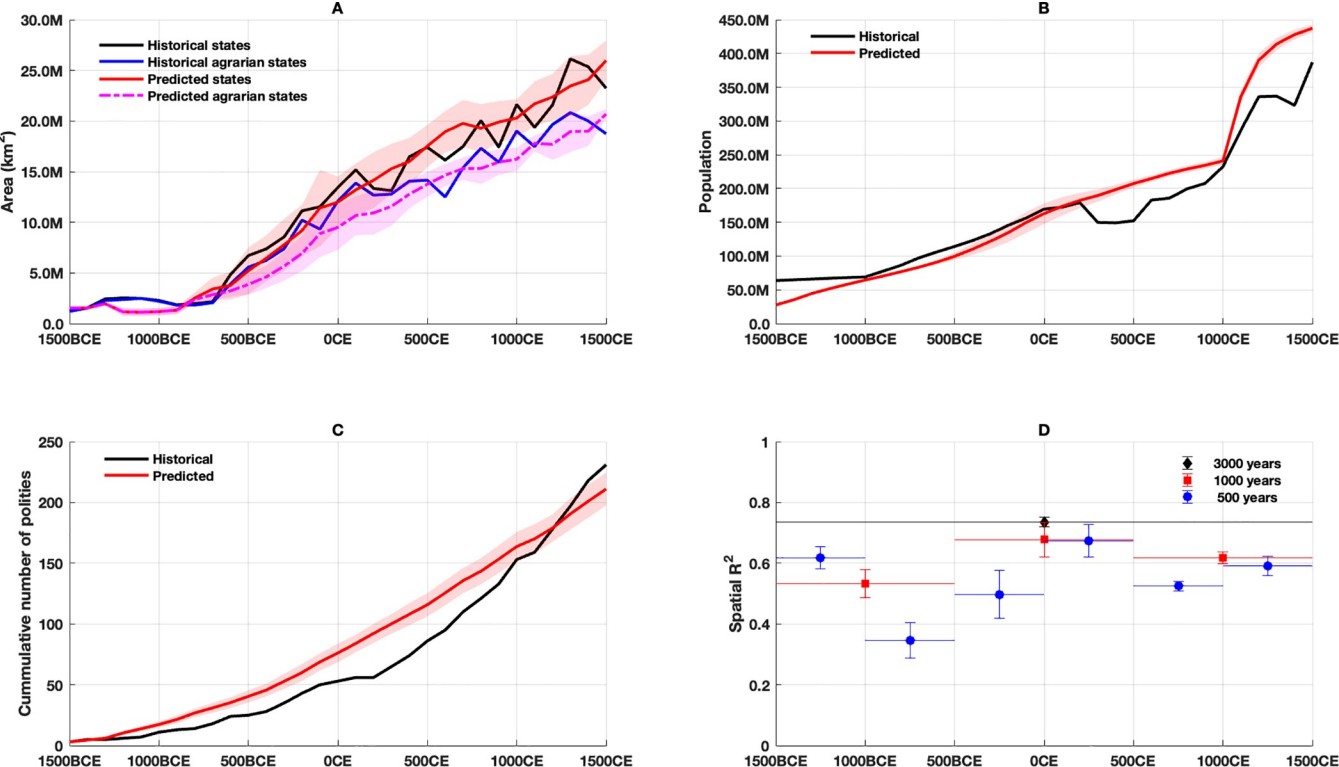

**Fig 4. Overall polity statistics.** Predictions for the Old World based on 32 trials (polities ≥ 10 regions). **A**: Predicted mean and standard deviation of total area under all large-scale polities (red) and agrarian states (magenta) compared with historical values per century from [16] (black and blue, respectively). **B**: Predicted mean and standard deviation of total population (red) compared with historical values every 20 years from KK10 (black). **C**: Predicted mean and standard deviation of cumulative (red) number of large-scale polities surviving 50 years compared with historical values (black) from [16]. **D**: Mean and standard deviation of spatial $R^2$ metrics for 500 (blue), 1000 (red) and 3000 (black) year intervals.

when growing agrarian populations trigger increased pressure from nomadic confederations, and when long-distance travel enables rapid expansion in the west.

While the population prediction (**Fig 4B**) largely tracks the KK10 historical population, the simple underlying demographic model does not address the significant historical plagues and famines in the mid-first and mid-second millennia CE or the effects of their rebound and, thus, the exogenous doubling of intensification by the model at 1000 CE causes the predicted population to slightly overshoot the historical values. Further, in spite of wide variance in agrarian state land occupation and associated carrying capacity, the variance in simulated overall population is small due to the lagged response of the logistical population model. The model produces a comparable number of long-lived (≥ 50 years, 2 'generations') polities (**Figs 4C** and **5**) as the historic record.

The spatial variance ($R^2$) results (**Fig 4D**) indicate that the predicted states also arise largely in the same regions as the historical data in different periods, especially after 500 BCE. Although each simulation starts with modeled agrarian states in roughly their historic locations, they spatially diverge rapidly from the historical expansion. This is reflected in the lower initial values of the 500-year metrics shown in **Fig 4D**. Nevertheless, after the rise of nomadic pressure, on the 1000-year (and 3000-year) scale the simulation does remarkably well. Overall on the 3000-year scale the model consistently explains 73% of the spatial variance of the historical data, an improvement over previous results of [16, 17]. Importantly, unlike those earlier

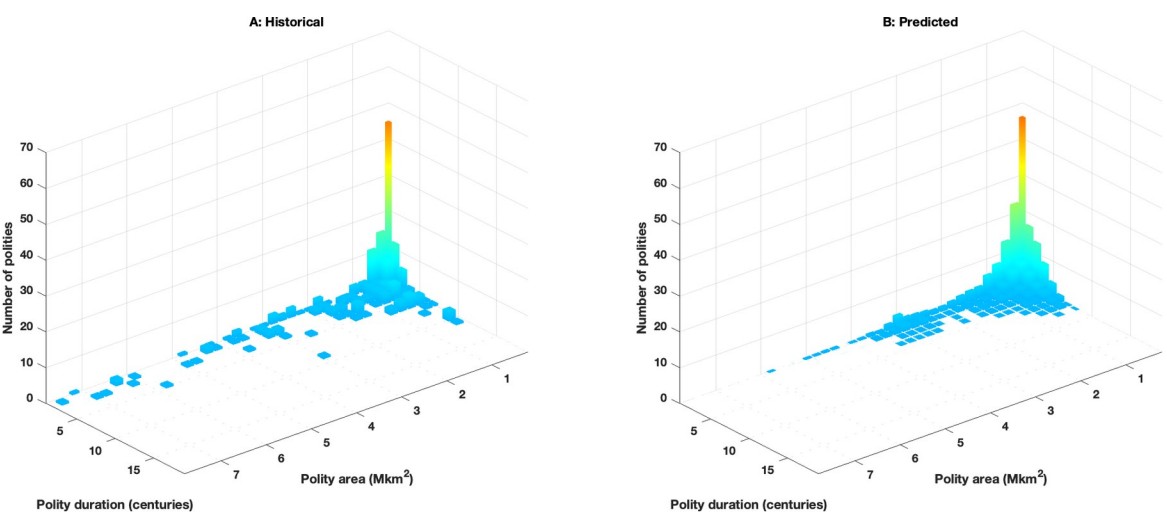

**Fig 5. Polity size and duration distribution.** Historical (**A**) and mean predicted (**B**) distributions of peak (agrarian and nomadic) duration (in centuries) and peak size (in millions (M) of km$^2$) between 1500 BCE and 1500 CE.

attempts, these estimates include predictions of the location and timing of nomadic confederations.

**Fig 5** compares the distribution of historical and predicted polities by duration (in centuries) and peak size (in millions of km$^2$). Since all modeled states employ the same parameters, if states remained independent of one another and were in similar geographical situations, their size and duration would be effectively identical. However, the influence of the underlying geography and the interaction between states (powerful or not) produce a wide variance in both the size and duration of states. The distribution of predicted states matches the historical record well except that the model predicts an excess of smaller states (corresponding to the increased cumulative number states in **Fig 4C** before 1000 CE) and somewhat under-estimates the number of very large states ($>$ 3 million km$^2$). Of course, the existence of modeled agrarian states beyond 3 million km$^2$ is permitted only because military efficiency increases under the influence of the steppe nomads; the relatively small number of very large agrarian states also reflects the few locations in Eurasia that can support these empires.

Strikingly, those very large states do not have correspondingly long lifetimes, as might be expected. In the model, this is largely a consequence of the underlying demographic-structural collapse mechanisms acting on all agrarian states. Simulations of small independent states in large agricultural plains and starting from a low population (as might happen at the beginning of the simulation), suggest they would all saturate their elite opportunity and collapse at roughly the same time, around 500 years ([17] S1 File). Subsequent states formed from those demographic-structural crises, however, last for only around 200 years because, while their base natural net birth rate does not change, their initial populations are larger so less time is required to approach saturation of elite opportunity. This shorter time period is in line with typical state 'secular cycle' times observed by [30]. Statistically I find that 30% of all states in nominal model runs succumb to demographic-structural collapse in this period; the rest are annexed by other, growing states. The odds of experiencing a demographic-structural crisis increases with the size of the state as, unsurprisingly, there are fewer rivals at larger scales that can successfully invade a large state. Indeed, all of the largest modeled empires succumb to demographic-structural collapse.

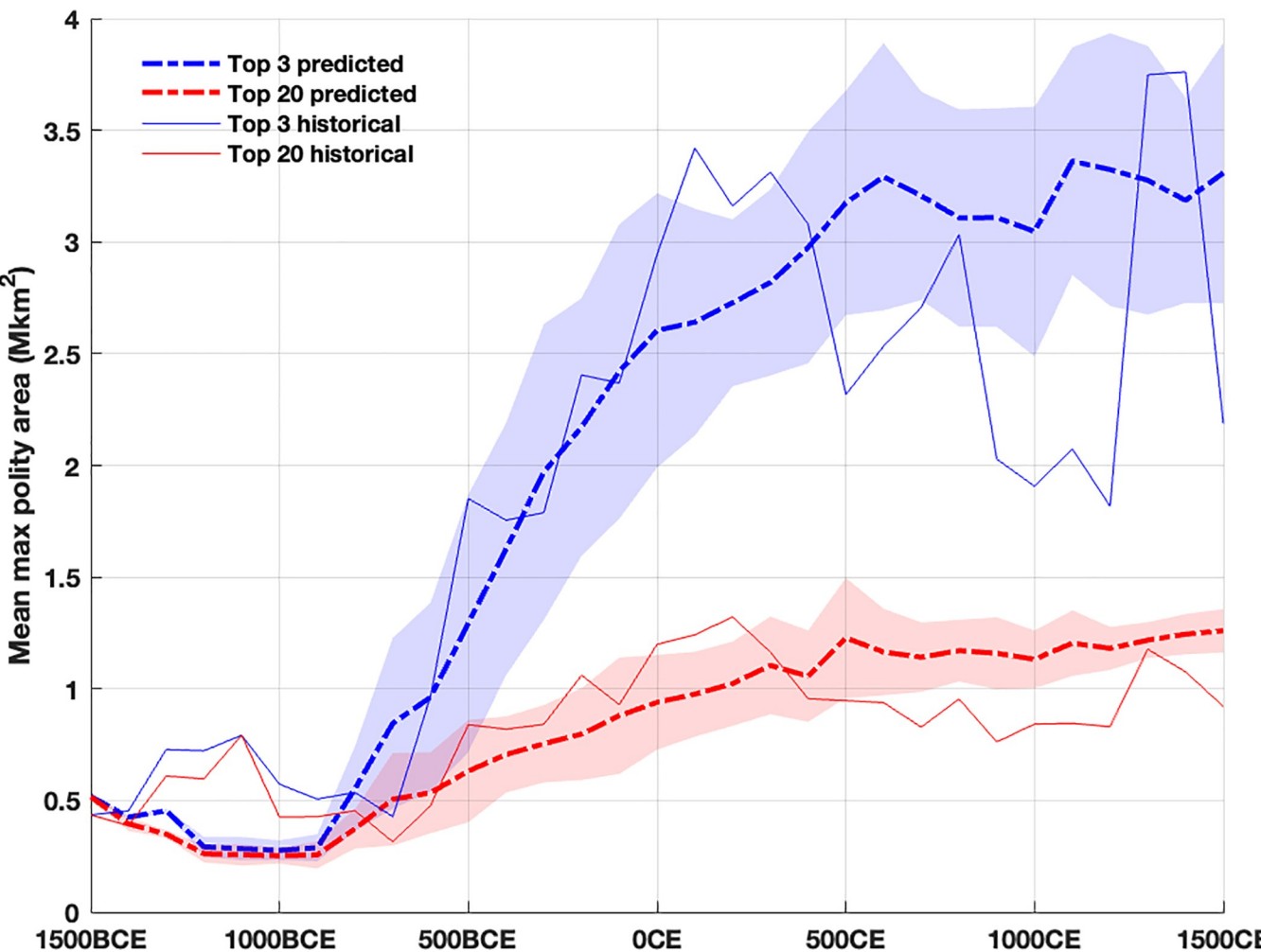

**Fig 6. Polity size trends.** Mean and standard deviation of predicted versus historical Old World polity sizes for the largest 3 and largest 20 states (agrarian and nomadic) between 1500 BCE and 1500 CE.

Expanding the size of a modeled state on its 'periphery' makes little difference to the rate at which its initial 'core' regions approach the crisis threshold. In the model, increased intra-state elite migration to annexed periphery can somewhat delay the onset of demographic-structural collapse (see **S1 File**) since it more equitably distributes elite opportunity and thus increases the time for any individual region to saturate its carrying capacity. However, to match the historical data on number of states, I find that the elite migration rate cannot have exceeded 0.01% per year suggesting that remaining in the core was an important elite behavior.

**Fig 6** shows the rapid historical and model-predicted increase in the sizes of the largest polities around 600 BCE, both quadrupling within 200 years. In the model the initial increase reflects the combined sizes of agrarian states in southwest Asia and their mirror nomadic confederations in the Pontic-Caspian steppe region. The temporal variance in the simulated increase of the these large polities occurs because it is possible for the (descendant) agrarian states from the initial states in the Fertile Crescent region to expand (stochastically) southward, away from the Pontic-Caspian region. While the expansion increases the population they remain distant from that region's strike zone, delaying the rise of nomadic confederations and

sometimes even permitting the first nomadic confederation to appear in east Asia several hundred years later; this pattern occurs roughly once every eight runs.

The top 3 very large modeled polities over all of Eurasia, while their location varies, continue to attain a mean size of around 3 million $km^2$ (roughly the size of the Roman Empire during the Principate, 0 CE). While the very large polities are few in each 100 year period, the mean size of the top 20 polities slowly doubles after 600 BCE to 1 million $km^2$ (roughly the size of the Byzantine Empire in 700 CE), again in accord with the historical trend. However, the model predicts the top 3 polities after 500 CE are consistently larger than their historical counterparts. This observation, coupled with the slight underestimate of the total number of states in **Fig 4C**, indicates that the model (incorrectly) predicts that the large basins of Europe, as well as southwest, south, and east Asia are all occupied by a smaller number of very large states that effectively 'quench' the formation of many smaller states observed in the historical record. The model thus makes all these theaters resemble east Asia with its recurrence of larger imperial regional powers, rather than the smaller, often competitive states that occupied Europe and parts of west and south Asia during Late Antiquity and the early Medieval period [10]. Clearly, other, differential behaviors must be incorporated into the model to account for the larger number of historical states in Europe [36] and south Asia after 500 CE.

## Counter-factual investigations

The systematic manipulation of the scale, timing and location of different event and technology contingencies provide counterfactual insights into the operation of the model and its ability to account for the historical record. I summarize several experiments here; see **S1 File** for further discussion.

The location, timing, range, and efficiency of long-distance military connections have a significant impact on how modeled agrarian states and overall military efficiency spreads in this period. Long-distance connections to distant agricultural regions expand the political border of a state beyond its strictly adjacent regions, increasing their annexation reach and the number of attack orders that they can write per time step. Thus, for a state with high military efficiency, occupying regions via long-distance connections can permit, all else equal, a more rapid and distant expansion of the state than a 'land-locked' state.

Inspired by [37], each explored long-distance connection type (sea, desert) is characterized exogenously, in different time periods, by a maximum distance that can be reached along a connection in a single time step and a transport 'efficiency', which reduces the effective distance used during power projection calculations. In the model state military efficiency is scaled by the *s* parameter which has units warriors/supply officer/100 km. Thus the distance discount factor has the effect of locally increasing the impact of *s*, by making, for example, 100 km to appear as 50 km under an efficiency of 0.5.

In the current model I limit sea-borne long-distance connections to littoral regions in the Mediterranean [16, 17] and desert-borne connections between 'oases' in West Asia and the Levant (see **S1 File**). I experimented with both efficiency and distance limits on both sea- and desert-transports, presumably using ships and, after 900 BCE, camels respectively. **Figs 7** and **8** show the logistical and 3000-year polity density impact, respectively, of different discount factors to these two different transport technologies.

Without either sea or desert travel (**Figs 7A** and **8A**) the state density in the western theaters is significantly reduced until after 500 CE while expansion in east Asia is unaffected. Disabling efficient desert transport but permitting efficient sea-borne transport effectively replicates the results of [17]: Thassolacratic state density in Europe is increased but no states from southwest Asia occupy India (**Figs 7B** and **8B**). Surprisingly, if sea travel is disabled but efficient desert

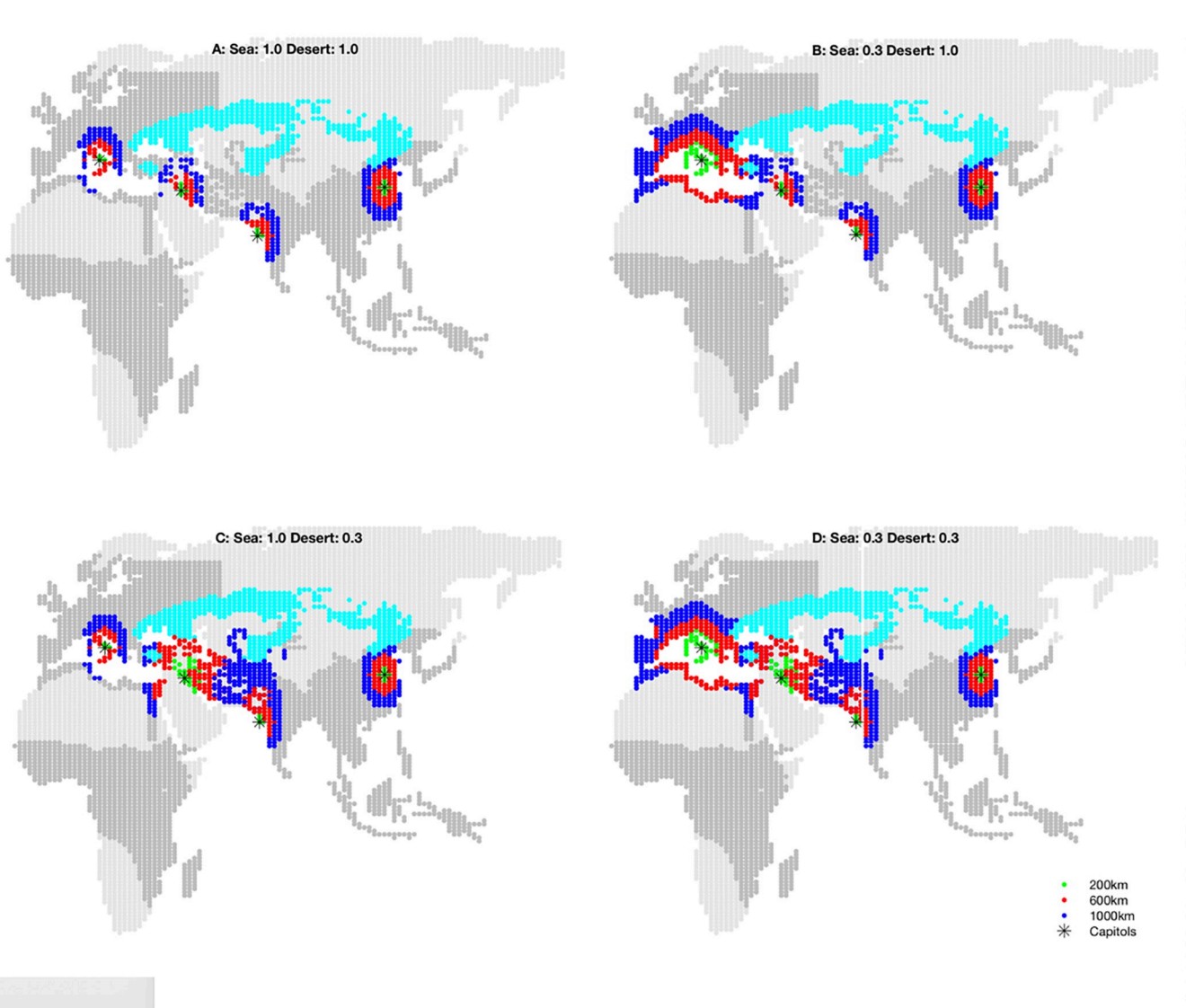

**Fig 7. Effective military distances.** Effective military distance to agricultural regions (dark grey) from four 'capitols' (Rome, Babylon, Mumbai, and Beijing) under different discount factors for long-distance sea and desert travel. Colors indicate how distant '200 km' (green) '600 km' (red) and '1000 km' (blue) appear for given a set of efficiency factors. Darker grey indicates agricultural regions; light blue indicates steppe. Map data republished from [16] under a CC BY license, with permission from Proceedings of the National Academy of Sciences, original copyright 2013. As Beijing enjoys no sea or desert travel advantage its distances remain unchanged in each panel. **A**: No travel advantage; the distances from the western capitals resemble Beijing's. **B**: Mediterranean sea travel discounted to 30% of normal overland travel; Rome is able to reach territories along the lip and deep into Europe; Babylon has some access. **C**: Near East desert travel discounted to 30% of normal overland travel allows Babylon and Mumbai to interact with one another if powerful enough. **D**: Combined discounted Mediterranean and Near East travel, used in the nominal simulation.

travel is enabled (**Figs 7C** and **8C**), the model suggests that the descendants of the very large agrarian states that arise in southwest Asia (under nomadic threat) expand more rapidly into south and then southeast Asia than in the nominal simulation (compare **Fig 8D**). This is due to the relative impediment to land expansion out of the narrow Levant, privileging the expansion of population and states to the southeast. Eventually, however, states in this counterfactual simulation expand into eastern Europe, spreading via the Balkans then north into Germany before spreading west into France and Spain after 500 CE.

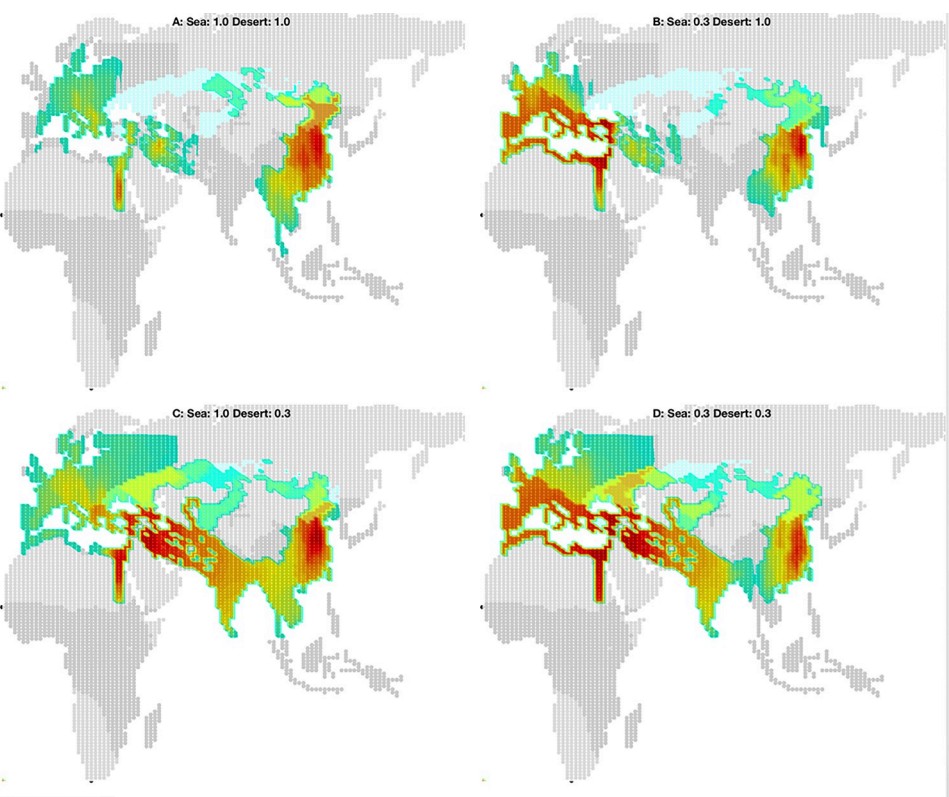

**Fig 8. Predicted polity densities given different effective military distances.** Corresponding impact over 32 trials on 3000-year predicted mean polity density of different discount factors applied to the western Eurasian sea and desert transport shown in **Fig 7**. Darker grey indicates agricultural regions; light blue indicates steppe. Panel D and polity density scale is identical to **Fig 3D**, right panel. Map data republished from [16] under a CC BY license, with permission from Proceedings of the National Academy of Sciences, original copyright 2013.

Efficient sea and desert travel in this western theater together form a key corridor of influence (**Figs 7D** and **8D**) where the unique geographical constraints were overcome early in the simulation period. Unless large, unified states roughly the size of the various Persian empires can arise in southwest Asia, enabled by these transports after the nomadic shock in 1000 BCE, the historical expansion of states in Europe and India cannot be accounted for by the model.

Delaying when efficient desert transport is adopted in the Near East has a strong regional impact. For example, if the camel were pressed into service 300 years later, in 600 BCE rather than in 900 BCE, the smaller, earlier states in the Near East are nevertheless pestered by nomadic tribes and, even with their slightly elevated $s$, they only slowly expand in the region. Although first nomadic confederations arise only a little later, around 500 BCE, the delay in efficient desert transport nevertheless retards the state expansion into Europe and India, reducing the imperial densities there. Developments in east Asia are, of course, unaffected. To ensure that a Pontic-Caspian confederation arises around 600 BCE under the late-camel assumption I find the confederation trigger population must be reduced to 4 million people. However, that lower threshold implies that when horse cavalry finally diffuses to east Asia, those agrarian states have often already exceeded this lowered threshold and a nomadic confederation arises immediately, 300 years prematurely.

Changing the timing of when initial states arise or when certain agricultural regions become productive also has an impact on the timing of nomadic confederations and regional imperial

density. For example, if the first east Asian agrarian states appear at the start of the simulation, in 1500 BCE rather than in 1200 BCE, the regional population in the tribal strike zone typically exceeds 6 million around 500 BCE, and once again an Asian nomadic confederation arises immediately and prematurely by 300 years. The assumed opening of the Pearl River valley in China to productive agriculture around 200 BCE (Fig 1A) constrains early Asian states to the Yellow and Yangtze river valleys, matching the early historical state densities in that region. A similar opening of northeastern European agricultural regions in late Antiquity (Fig 1B) constrain earlier European states to the north lip of the Mediterranean and western Europe. Premature opening of these areas instead allows early and substantial expansion of a few very powerful states into these larger regions. Thus, delayed opening of large areas of additional hinterland 'frontier' provides an important late asymmetric advantage that a few powerful well-situated states can exploit and permits an improved match to the historical record.

Disabling the nomadic threat completely impairs the model's ability to raise very large states; states continue to fill the agrarian theaters but, even with discounted sea and desert travel in the west, the expansion is very slow and ahistorical. One could argue that, even in the absence of nomadic pressures, agrarian states would hardly be idle and, perhaps following Kremer [38], would endogenously improve their military efficiency at a rate proportional to the Old World population. To investigate this alternative I implemented a simple 'Kremer mechanism' that exogenously doubles $s$ in 3000 years assuming a constant improvement rate that depends on the Old World population as Kremer assumed. Using KK10 population values I empirically estimated that rate to be roughly $1.3 \times 10^{-12}$ military innovations per person per year (see **S1 File**). However, since the starting population of the Old World is small, so are the early changes to $s$. In spite of the growing population and slow improvement to $s$ overall, Eurasia eventually becomes saturated by fairly large states but all are equivalently powerful (they all benefit from the improvements invented everywhere according to Kremer's model), inhibiting the dominance of very large states altogether. Thus the asymmetric and substantial military shock that emanates from the steppe early in the simulation appears required by the model to match historical events, as Turchin conjectured. This result is consistent with a recent statistical analysis [35] of comprehensive world-wide data on the evolution of military technology that identified several significant factors beyond world population and existing technology levels, notably the adoption of iron and horse cavalry and connectivity between states, which are required to causally account for the regional increase of historical military technologies.

Manipulation of the location and timing of the invention of horse cavalry impacts the formation of modeled nomadic confederations and large-scale agrarian states. For example, inventing horse cavalry in 1000 BCE as usual but shifting the longitudinal location of its invention away from the Pontic-Caspian steppe region to nearer the east Asian steppe border permits a large Asian state and nomadic confederation pair to arise only somewhat earlier, around 400 BCE rather 200 BCE, in a manner similar to the southwest Asian pair in the nominal simulation. In this case, while horse cavalry diffusion is no longer an issue, 600 years are nevertheless required for the Asian states, beginning in 1200 BCE, to reach the confederation triggering population, even under increased pressure from nomadic tribes. Further, while nomadic confederations eventually appear in the west, no corresponding, large-scale southwest Asian agrarian states arise because so many smaller states have already saturated that region and, unlike east Asia, there remains little hinterland there to permit one state to expand at the expense of another. However, large states do eventually form in the distant European and Indian subcontinent theaters as states in those (still peripheral) theaters exploit the advantages of higher $s$ (attained via propagation as usual) to conquer that remaining hinterland.

As suggested above, increasing or decreasing the confederation trigger population in the nominal simulation delays or advances the time when confederations occupy the steppe and, thus, when very large agrarian states could form. In a similar fashion, one might argue that horse cavalry itself was invented by nomadic tribes *in response to* increasing agrarian wealth near the steppe itself. Again, using predicted population as a proxy for wealth, during nominal runs I find the combined population of the agrarian states within the Pontic Caspian tribal strike zone in 1000 BCE to be roughly 3 million people. Running simulations with 3 million people in any tribal strike zone as a trigger for the invention of horse cavalry finds it arising around 1100 BCE (± 100 years), largely to the west of the Pontic Caspian region but infrequently in Asia (1 out of 32 runs). This result depends both on the late appearance of the first Asian state in 1200 BCE and the stochastic development of Near Eastern states away from the western strike zones and the stochastic development of Asian states toward the eastern strike zone. If, instead, the first Asian state arose earlier, say in 1500 BCE, the 3 million invention trigger predicts that horse cavalry would have been consistently invented east of the Altai mountains, rather than in the Pontic Caspian region, *circa* 1300 BCE, three hundred years earlier and 5000 km away.

Of course other factors in addition to the availability of substantial but vulnerable agrarian populations may have encouraged the invention of horse cavalry in the Pontic-Caspian region. Two resources unique to that region suggest themselves. First, it may have required access to superior horses such as the "heavenly horses that sweat blood" so prized by the Han emperors and located in the Fergana Valley in Transoxiana. Second, one of the key materials used in the manufacture of the powerful nomadic compound bow is *isinglass*, a glue made from the swim bladders of fish. The numerous rivers emptying into the Black and Caspian Seas, but not present in quantity in China, may have supplied this critical glue. Assembling a bow took significant expertise and required a year to construct. This long manufacturing time coupled with rare ingredients may go some distance to explain the relatively long diffusion time of nomadic horse cavalry tactics across the steppe compared with the otherwise apparently rapid increases in military efficiencies and weaponry by agrarian states.

Together these counter-factual experiments demonstrate how different contingencies interlock to constrain the emergent dynamic behavior of modeled states. The nominal contingencies employed here are necessary and, taken together with the underlying model behavior, are sufficient to account for the historical dynamics of states in this period.

## Discussion

The underlying model behavior suggests that the history of Old World agrarian states is not simply 'one damn thing after another', but rather two: Interstate wars that fuel growth by conquest followed, after a time, by a civil war that fractures the state, triggered by structural limits to that growth. This 'rhyming' process among the warrior elite—alternately cooperating against competitive external threats, then competing with each other, repetitive but not cyclic —appears to account for the spread of very large agrarian states for thousands of years in Eurasia.

Critically, during this period, these two dominant, driving 'damnations' were accompanied by a third: a sequence of powerful spatially and temporally asymmetric extortionary threats from steppe nomadic confederations that incentivized increased military efficiency and triggered a cascade of larger, more powerful agrarian states. As Turchin [23] conjectured, the model shows that this long-term military asymmetry on the ecological border between steppe and agriculture can, in fact, lead to the sequenced rise of very large, stable "mirror" states.

Similar long-term military asymmetries appear to drive major shifts in state sizes and boundaries throughout history [39]. Notably, beginning in 1500 CE, the refinement and deployment at scale of gunpowder weapons on ocean-going vessels again 'changed the meaning of geography' [9] and permitted European states to rapidly colonize vast swathes of Africa, Polynesia, and the Americas, forming extremely large empires. Gunpowder weapons on horseback allowed Russia to match and subdue the steppe nomads over the next few hundred years. Finally, if typhoid and smallpox are considered an asymmetric military advantage, however un-intentional, these diseases, in addition to horse cavalry and gunpowder weapons, facilitated the rapid European conquest of the Americas.

Nevertheless, in spite of their increasing ability to annex more territory after 600 BCE, agrarian states failed to escape repeated demographic-structural crises of intra-elite competition over dwindling spoils. Historically, such fractious crises have tended to result in intense civil conflict and, at times, societal collapse. There is little evidence that the states of this period were able to ward off such crises for long through policies that might have reduced elite competition or invested in productivity gains that expanded the spoils to be shared.

Work on early modern [19, 30] and modern states [40–42] suggest that even states that do finally, for example, increase their agricultural production, establish constitutional term-limits for elite leaders, expand their economies into other (technologically-driven) sectors, and engage in substantial international trade continue to face structural limits to growth that trigger political and social (elite) instability. These analyses suggest that demographic-structural crises emerge slowly over many (~6–8) generations before rapidly becoming acute, which may make them especially difficult for societies to avert politically, even if they recognize their recurrence and vary in their resilience to the upheaval. Future work is needed to explore what sort of interventions could be made at scale and how counterfactual histories might have unfolded with states that could somehow avoid such demographic-structural crises.

Historians will hardly be surprised about the type of forces and behaviors employed in the model. They should be surprised, however, at the relative economy of means that appears sufficient to substantially and quantitatively account for the historical observations about state spread in the Old World. The model largely emphasizes the military aspects of state behavior for both agrarian states and nomadic confederations. While the historical literature and scholarly debates identify much more complex and nuanced behaviors engaged in by both types of polities, the model results here suggest that these other factors and social forces, notably trade, specific governance and institutional arrangements, or cultural-ideological packages play apparently secondary roles during this period—they are unneeded in the model to explain much of the observed variance in historical state formation. This is unlikely to remain the case as additional datasets at more refined spatial and temporal scale are gathered. These cultural and institutional factors, which buttress the gains of conquest, are likely to play a larger role in explaining why the current model fails to account for the historical record in certain areas. Further, these factors will likely be key to modeling the spread of language and religion, bureaucratic scale, and social and wealth inequality. Yet even here, work by Turchin *et al.* [43] observing a common trajectory of social complexity developments as population scale and polity size increase for hundreds of world-wide polities at different times, suggests that simple models of the more detailed interior working of societies may be expected.

While far from the final word, mechanistic dynamic models such as the one presented here demonstrate the potential to unify historical arguments, constrain plausible causal paths (via counter-factual experiments), quantitatively bound the relative impact of additional or absent behaviors, and identify required technological scales and geographical boundary conditions, necessary and sufficient to account for observed history. History, at least at this level of analysis, appears to be subject to a few powerful mechanisms of substantial scope, most already

outlined by historians and sociologists, but now testable against comprehensive well-sampled data and subject to scientific investigation.

## Supporting information

**S1 File.**
(PDF)

**S1 Video. Example model simulation.**
(MP4)

## Acknowledgments

I thank Dan Hoyer, Stan Lanning, Fritz Stahr, Peter Turchin, and Kai Wirtz for many important discussions. Walter Scheidel and Ian Morris were generous with their incredulity and encouragement. I also thank the two reviewers for encouraging several important improvements and clarifications.

## Author Contributions

**Conceptualization:** James S. Bennett.

**Formal analysis:** James S. Bennett.

**Investigation:** James S. Bennett.

**Methodology:** James S. Bennett.

**Software:** James S. Bennett.

**Validation:** James S. Bennett.

**Visualization:** James S. Bennett.

**Writing – original draft:** James S. Bennett.

**Writing – review & editing:** James S. Bennett.

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
