## [Decision Letter · Decision Letter 0]

31 May 2021

PONE-D-21-12450

Retrodicting the rise, spread, and fall of large-scale states in the Old World

PLOS ONE

Dear Dr. Bennett,

Thank you for submitting your manuscript to PLOS ONE. After careful consideration, we feel that it has merit but does not fully meet PLOS ONE’s publication criteria as it currently stands. Therefore, we invite you to submit a revised version of the manuscript that addresses the points raised during the review process.

All comments need to be addressed before re-submission.

We look forward to receiving your revised manuscript.

Kind regards,

Peter F. Biehl, PhD

Academic Editor

PLOS ONE

Additional Editor Comments:

Your manuscript has now been seen by two referees, whose comments are appended below. You will see from these comments that while the referees find your work of potential interest, they have raised substantial concerns that must be addressed. In light of these comments, we cannot accept the manuscript for publication, but would be interested in considering a revised version that addresses these serious concerns.

We hope you will find the referees' comments useful as you decide how to proceed. Should presentation of further data and analysis allow you to address these criticisms, we would be happy to look at a substantially revised manuscript. However, please bear in mind that we will be reluctant to approach the referees again in the absence of major revisions.

Journal Requirements:

2. We note that Figures 1, 5, 6, 7, 9, S2, S4, S8, S11, S13, S16, S21, S23, S26, S28 and S2 Animation in your submission contain map images which may be copyrighted. All PLOS content is published under the Creative Commons Attribution License (CC BY 4.0), which means that the manuscript, images, and Supporting Information files will be freely available online, and any third party is permitted to access, download, copy, distribute, and use these materials in any way, even commercially, with proper attribution. For these reasons, we cannot publish previously copyrighted maps or satellite images created using proprietary data, such as Google software (Google Maps, Street View, and Earth). For more information, see our copyright guidelines: http://journals.plos.org/plosone/s/licenses-and-copyright.

You may seek permission from the original copyright holder of Figures 1, 5, 6, 7, 9, S2, S4, S8, S11, S13, S16, S21, S23, S26, S28 and S2 Animation to publish the content specifically under the CC BY 4.0 license. 

If you are unable to obtain permission from the original copyright holder to publish these figures under the CC BY 4.0 license or if the copyright holder’s requirements are incompatible with the CC BY 4.0 license, please either i) remove the figure or ii) supply a replacement figure that complies with the CC BY 4.0 license. Please check copyright information on all replacement figures and update the figure caption with source information. If applicable, please specify in the figure caption text when a figure is similar but not identical to the original image and is therefore for illustrative purposes only.

Reviewers' comments:

Reviewer's Responses to Questions

**Comments to the Author**

1. Is the manuscript technically sound, and do the data support the conclusions?

Reviewer #1: Yes

Reviewer #2: Partly

2. Has the statistical analysis been performed appropriately and rigorously? 

Reviewer #1: Yes

Reviewer #2: Yes

3. Have the authors made all data underlying the findings in their manuscript fully available?

Reviewer #1: Yes

Reviewer #2: Yes

4. Is the manuscript presented in an intelligible fashion and written in standard English?

Reviewer #1: Yes

Reviewer #2: Yes

5. Review Comments to the Author

Reviewer #1: This paper represents a significant step towards understanding the limited set of factors driving large-scale repetitive patterns in the long-term socio-political development in the “Old World”. Model outcomes correspond to a significant number of values obtained from historical data. The disagreements between modeled and historical data are discussed in the text. Model assumptions are clearly presented. I suggest just a few minor improvements clarifying the text and slightly extending the discussion of the model outcomes. These are as follows:

1. Discussing the increase in military efficiency of steppe populations c. 1000 BCE (vs. 1500 BCE) it might be reasonable to specify that nomadisim as a form of economic organization and the lifeway in steppe forms in the western part of the Great Eurasian Steppe in the very end of the 2nd mil BCE – the very beginning of the 1st mil. BCE (e.g., Khazanov 1984; Bunyatyan 2003). Leaving aside the issues of earlier origin of horse riding and military conflicts between Eneolithic and Bronze Age semi-nomads and agriculturalists (e.g. Anthony 2007), Khazanov’s conclusion that the Bronze Age semi-nomads did not seem to be a real threat to Early States supports the model’s assumptions (pp. 4-5) and outcomes. Khazanov’s book also discusses the appearance of nomads in the eastern part of the Great Eurasian Steppe c. in the middle of the 1st mil BCE, which finds an agreement with the results of simulations.

2. From the perspective of archaeological record, localization of the origin of “military cavalry” between the Black Sea and the Caspian Sea raises doubts. Synchronous horse riding and arrowheads, short swords and daggers are evident across much wider territory – from the North Pontic region to the Volga region. Maybe, it would be more reasonable to emphasize the role of military tactics already assumed by the author. This would provide a stronger ground for localization of the invention and formation of the nomads federation in the smaller region.

3. Sarmatian time (c. 4th/3rd century BCE – 4th century BCE) in the western part of the Great Eurasian Steppe is associated with the invention of stirrup causing the appearance of heavily equipped riders known in this area from the 1st century BCE (e.g., Simonenko 2015 and earlier papers by Khazanov). Huns (4th – 6th century BCE) were equipped by complex compound bows. I would briefly discuss how these military inventions in steppe fit the model outcomes.

4. It seems to be reasonable to add a few sentences summarizing the outcomes of Bennett 2016 paper in the introduction (p. 5, line 94). Otherwise, reading may be somewhat biased from all the model assumptions and parameters towards the arm race between two types of polities.

5. Since the model starts at 1500 BCE, maybe it would be reasonable to replace the “Middle Bronze Age” by the “Late Bronze Age” according to the Bronze Age periodization developed for the Near East and the western part of the Great Eurasian Steppe? I am not familiar with the periodization of the eastern part of steppe – if this date corresponds to the Middle Bronze Age there, maybe “Middle/Late Bronze Age”?

6. Figure 4 in the text is referenced before Figure 1 (p. 3, line 64). I would remove this reference in order to keep the ordering of figures and references to them.

7. P. 17, line 373, “only” – please replace the semicolon by “l”.

Anthony, D.W. (2007). The Horse, the Wheel and Language. How Bronze-Age Riders from Eurasian Steppe Shaped the Modern World. Princeton: Princeton University Press.

Bunyatyan, K.P. (2003). Correlations between agriculture and pastoralism in the Northern Pontic Steppe area during the Bronze Age. In: Levine, M., Renfrew, C. and Boyle, K. (eds.), Prehistoric Steppe Adaptation and the Horse. Cambridge: McDonald Institute for Archaeological Research, 269-286.

Khazanov, A. (1984). Nomads and the Outside World. Cambridge: Cambridge University Press.

Simonenko, A. (2015). Sarmatskie vsadniki Severnogo Prichenomorya [2nd edition]. Kiev: Oleh Filiuk. [English transliteration of the Russian title].

Reviewer #2: ## Summary and recommendation

This work attempts to retrodict the trajectories of large states in Afro-Eurasia between 1500 BCE and 1500 CE, building over the work of Turchin et al. on macro-scale modeling. The author presents a new version of Turchin et al. model where there are extra contingencies to specific mechanisms and, most significantly, where steppe nomadic confederations are implemented differently from agrarian states.

The paper is focused on presenting, interpreting, and discussing simulation results, while essential elements of model description are somewhat lacking or only found in Supplementary Information files. Unfortunately, the manuscript is also excessively long and not well-structured, which delays proper comprehension and assessment. The author has invested significant work in developing the model, running experiments, and analyzing results; however, the delivery of contents, including text and figures, and its contextualization within historical investigation does not make justice to this effort.

I believe this work is worth publishing and that it has the potential to be an important contribution, relevant to a broad audience. However, I recommend its publication only after addressing the following points.

## Comments on Manuscript

*Content*.

Point 1. The manuscript covers several aspects that either follow, extend, or revise Turchin et al. and their sources. The author is clearly aligned with a general model of Eurasian "cliodynamics" based on a clear-cut separation (and political and military opposition) between so-called agricultural and nomadic polities. This position is entirely valid. However, the author should at least mention (and potentially discuss) a very different perspective, quite extended among historians and archaeologists. This other line of thought considers that the degree and nature of this separation have been exaggerated by biased discourses, encouraged by contemporary and later polities, up to modern nation-states. I highly recommend taking into consideration the work of Anatoli Khazanov (e.g., his inaugural book, "Nomads and the Outside World", 1984). More generally, the manuscript would benefit from acknowledging the nuances indicated by many historians and archaeologists working in Afro-Eurasia. For example, specifically on Central Asia, see Michael Frachetti's work on the so-called Inner Asia Corridor, Christoph Baumer's book "The History of Central Asia", and UNESCO's "History of Civilizations of Central Asia".

- Frachetti MD, Smith CE, Traub CM, Williams T. 2017 Nomadic ecology shaped the highland geography of Asia's Silk Roads. Nature 543, 193–198. (doi:10.1038/nature21696)

- Baumer C. 2012 The History of Central Asia: Volume 1: The Age of the Steppe Warriors. I. B. Tauris.

- Asimov MS, Bosworth CE, editors. 1998 History of Civilizations of Central Asia, Volume IV, The age of achievement: A.D. 750 to the end of the fifteenth century, Part One: The historical, social and economic setting. UNESCO Publishing. (the entire collection can be found here: https://en.unesco.org/themes/generalregionalhistories#centralasia)

Point 2. There is a general over-explanation of results in qualitative terms while lacking details about quantitative patterns and the more systematic comparisons between simulations and in relation to the historical data. The tone in Results dangerously confounds simulations and history, adding too much ad hoc interpretation and skipping important model details, which are left somewhat disconnected in S1, which in turn mixes model description and results. Simulation run "chronicles" can be helpful to illustrate the model common behaviors and even raise insight but are a limited resource in terms of drawing theoretical generalizations. I understand the challenge of dealing with simulations that take up to 10 minutes and output large amounts of spatially distributed data. Still, fitting results to the historical data must come together with a clear explanation of how the mechanisms programmed affect aggregated results, which is partially already done in S1 but not so much in the main text.

*Format*.

Point 1. The paper is fundamentally clear but excessively long and wordy, sometimes even repetitive. Respecting the author's style, I would recommend at least re-visiting the general structure of the text and re-assessing what is explained in each paragraph and how (e.g., summarizing them as "titles" and checking if their content connects well with the rest of the flow).

Point 2. As I understand, PLOS ONE format does not place Materials and Methods at the end of research papers, so I highly recommend this section to be moved to its proper place after Introduction. This note is vital for an article dealing with a simulation model. In this case, the "methods" are not protocols applied to empirical data in order to obtain results (i.e., a mean to an end) but contains in itself the core of the contribution, which is mainly a theoretical and methodological proposition. For readers, it is not possible to evaluate results or even fully understand their implications if a minimum overview of the model does not precede them.

Point 3. There are few and scattered mentions of model assumptions and "hard-coded" behaviors. I strongly suggest adding a bullet-point list, table, or diagram showing the contingencies that are pre-programmed by "deus_ex_machina()".

Point 4. Also, the manuscript could benefit from re-evaluating the role of figures and their order. I strongly recommend that figures are numbered following their mention in the text as I believe it is a general editorial policy. For instance, Fig.4 is the first figure mentioned already in the Introduction. Is this mention necessary? If so, would it not be a better idea to represent the historical data considered in a way that is clearly differentiated from simulation results (as a new Fig. 1)?

Point 5. Regarding the main form of visualization of results (maps in Fig. 1, 5, 6, etc.), I recommend that notations and the meaning of colors be shown directly on the figure (i.e., titles, legend, etc.), particularly in the case of Fig. 1.

Point 6. Figures 8 and 10 are too abstract and crude and do not improve the reader's understanding of the model. I would recommend searching for alternative types of diagrams and infographics, including at least some text in the figure.

---

## Comments on Code and code documentation

Please publish the content of "S3 Code.zip" as a repository, following the FAIR principles, including making it citable. This is paramount for upholding the value of the model and the article in the long run. Offering software as Supplementary Information to the article is not a good option in general, but here it is particularly unacceptable given the importance, size, and complexity of the content. I would personally also include S1 and S2 to S3 together in a separate public and published repository. The materials can then be cited using DOI referring to stable repository servers.

I recognize the effort to explain the code in "README.md", but I would strongly recommend following a more visual and structured approach (i.e., sections and subsections, bullet points, tables, figures). Furthermore, I believe much of what is explained in S1 should be present here as the model documentation, including some direct demonstrations of the mechanisms implemented in each method. In any case, this should not be kept in S1, which passed a few years may well become inaccessible or unfindable.

I ask the author to be more precise when naming methods and variables and closely follow the single responsibility principle. The current names are amusing and understandable enough after further investigation. Still, the entire code would become easier to read and understand if the relationship between name and content/meaning would be more precise. Most methods are also quite long and complex (e.g., deus_ex_machina()) and could benefit from being broken down into smaller methods (i.e., refactoring).

6. PLOS authors have the option to publish the peer review history of their article (what does this mean?). If published, this will include your full peer review and any attached files.

Reviewer #1: No

Reviewer #2: **Yes: **Andreas Angourakis

---

## [Decision Letter · Decision Letter 1]

9 Nov 2021

PONE-D-21-12450R1Retrodicting the rise, spread, and fall of large-scale states in the Old WorldPLOS ONE

Dear Dr. Bennett,

Thank you for submitting your manuscript to PLOS ONE. After careful consideration, we feel that it has merit but does not fully meet PLOS ONE’s publication criteria as it currently stands. Therefore, we invite you to submit a revised version of the manuscript that addresses the points raised during the review process.

We look forward to receiving your revised manuscript.

Kind regards,

Peter F. Biehl, PhD

Academic Editor

PLOS ONE

Journal Requirements:

Additional Editor Comments (if provided):

Please address comments by reviewer 2 in detail.

Reviewers' comments:

Reviewer's Responses to Questions

**Comments to the Author**

1. If the authors have adequately addressed your comments raised in a previous round of review and you feel that this manuscript is now acceptable for publication, you may indicate that here to bypass the “Comments to the Author” section, enter your conflict of interest statement in the “Confidential to Editor” section, and submit your "Accept" recommendation.

Reviewer #1: All comments have been addressed

Reviewer #2: All comments have been addressed

2. Is the manuscript technically sound, and do the data support the conclusions?

Reviewer #1: Yes

Reviewer #2: Yes

3. Has the statistical analysis been performed appropriately and rigorously? 

Reviewer #1: Yes

Reviewer #2: Yes

4. Have the authors made all data underlying the findings in their manuscript fully available?

Reviewer #1: Yes

Reviewer #2: Yes

5. Is the manuscript presented in an intelligible fashion and written in standard English?

Reviewer #1: Yes

Reviewer #2: Yes

6. Review Comments to the Author

Reviewer #1: The author addressed all comments from the previous round of reviewing. I recommend this interesting and important study for publication.

Reviewer #2: As a general, rather personal note, I suggest that the author rethink one last time the balance of content between the main manuscript and S1. The text in the manuscript is excessively indirect and narrative, explaining model characteristics and behaviours in a rather vague and wordy style. Almost no reference to the model implementation, experimental setting and output analysis is made before jumping into results. In contrast, the text in S1 explains quite clearly some critical points about the model and its results. Additionally, as it is, S1 is almost a second paper, yet grossly repeating several points already made in the main text. As a reader and someone that also use simulation in this field, I would insist, at least, that the author brings more of S1 content to the main manuscript, at his discretion.

This is, however, aimed at raising the quality of the final text, and not a call-out for a third review round.

# minor suggestions about format

- lines 89, 426, 658: a-historical -> ahistorical

- lines 150, 155, 165, 512: with "net birthrate", "net effective birthrate", or "birth rate", I believe the author means the "population natural growth" or "population natural change", that is, population change that accounts for births and deaths but not migration flows. E.g., see: https://ec.europa.eu/eurostat/statistics-explained/index.php?title=Glossary:Natural_population_change

- lines 176-177, Fig 1 caption: "Steppe shown in light blue" is not needed since this is already clear in the legend.

- line 344: each another -> each other

- Fig 3, lines 384-386, and Fig 3 caption: Fig 3 and its description is still quite confusing while also repetitive. The titles added to the figure are not intuitive ("1000 years ending 500BCE"), and the panel reference as left-right and rows as letters only make things more difficult to interpret. I suggest modifying this figure by adding short concise titles for each row and column, treating it as a 4x2 table, and saving the reader the trouble of deciphering a detailed description in the text. As I see it, these would be: "1500-500BCE", "500BCE-500CE", "500CE-1500CE", "1500BCE-1500CE", for rows; "historical" and "simulated", for columns. If PLOS editorial requires numbering each subplot, I would add letters for each one and not only for rows (e.g., A to H, or A.1 to D.2), but would still have row and columns titles. Last, please make sure that all text is large enough to be legible.

- Fig 4: this figure is really four different plots unless they are reordered so that they share the same x-axis (time). Of course, this would only make sense if the author intends to call the reader's attention to trends in each variable at specific periods. In any case, it would make it much easier to read if legends were included.

- Fig 7 and 8 captions: these are unnecessarily long and describe information that can be better shown directly on the figure. There is no need to explain colors if there is a clear legend title and labels. I would suggest the same approach I recommend for Fig 3, i.e., adding clear and concise titles per row and column.

- Regarding the Zenodo repository: please make sure to add a reference to the paper once published and that the paper refers to the DOI of the latest version or ALL versions. Also, I recommend adding a short description of the repository in Zenodo -- even if only a sentence or two.

Congratulations and best of luck!

7. PLOS authors have the option to publish the peer review history of their article (what does this mean?). If published, this will include your full peer review and any attached files.

Reviewer #1: No

Reviewer #2: **Yes: **Andreas Angourakis

---

## [Author Response · Author response to Decision Letter 1]

1 Dec 2021

See detailed comments in the uploaded Response to reviewers document.

---

## [Editor Report · Decision Letter 2]

13 Dec 2021

Retrodicting the rise, spread, and fall of large-scale states in the Old World

PONE-D-21-12450R2

Dear Dr. Bennett,

We’re pleased to inform you that your manuscript has been judged scientifically suitable for publication and will be formally accepted for publication once it meets all outstanding technical requirements.

Kind regards,

Peter F. Biehl, PhD

Academic Editor

PLOS ONE
---

## [Editor Report · Acceptance letter]

17 Dec 2021

PONE-D-21-12450R2 

Retrodicting the rise, spread, and fall of large-scale states in the Old World 

Dear Dr. Bennett:

I'm pleased to inform you that your manuscript has been deemed suitable for publication in PLOS ONE. Congratulations! Your manuscript is now with our production department. 

Kind regards, 

on behalf of

Dr. Peter F. Biehl 

Academic Editor

PLOS ONE